# Direct observation of altermagnetic band splitting in CrSb thin films

Sonka Reimers [1], Lukas Odenbreit[1], Libor Šmejkal [1,2], Vladimir N. Strocov[3], Procopios Constantinou [3], Anna B. Hellenes [1], Rodrigo Jaeschke Ubiergo [1], Warlley H. Campos [1], Venkata K. Bharadwaj[1], Atasi Chakraborty [1], Thibaud Denneulin [4], Wen Shi[4], Rafal E. Dunin-Borkowski [4], Suvadip Das [5,6], Mathias Kläui [1,7], Jairo Sinova[1,8] & Martin Jourdan [1] ✉

Altermagnetism represents an emergent collinear magnetic phase with compensated order and an unconventional alternating even-parity wave spin order in the non-relativistic band structure. We investigate directly this unconventional band splitting near the Fermi energy through spin-integrated soft X-ray angular resolved photoemission spectroscopy. The experimentally obtained angle-dependent photoemission intensity, acquired from epitaxial thin films of the predicted altermagnet CrSb, demonstrates robust agreement with the corresponding band structure calculations. In particular, we observe the distinctive splitting of an electronic band on a low-symmetry path in the Brilliouin zone that connects two points featuring symmetry-induced degeneracy. The measured large magnitude of the spin splitting of approximately 0.6 eV and the position of the band just below the Fermi energy underscores the significance of altermagnets for spintronics based on robust broken time reversal symmetry responses arising from exchange energy scales, akin to ferromagnets, while remaining insensitive to external magnetic fields and possessing THz dynamics, akin to antiferromagnets.

Altermagnets (AMs) represent a new recently recognized class of magnetically ordered materials with a collinear alignment of identical magnetic moments, which exhibit an alternating spin polarization pattern within the electronic bands in reciprocal space[1–5]. This exceptional property has sparked substantial interest in exploring innovative applications in the realm of spintronics. The goal is to harness the combined advantages presented by antiferromagnets (AFs), encompassing ultrafast dynamics[6] and resilience in the presence of external magnetic fields[7], all while capitalizing on the substantial transport and optical effects[8–10] conventionally ascribed solely to spin-polarized currents and strongly spin-split bands observed in ferromagnets and

ferrimagnets. Consequently, these materials have the potential to lay the foundation for groundbreaking device concepts.

In a manner akin to conventional collinear antiferromagnets, AMs are comprised of sublattices that exhibit ferromagnetic ordering in each sublattice, but an antiparallel inter-sublattice alignment, leading to compensated collinear order. As with antiferromagnets, this compensated order is protected by a symmetry that interchanges the antiparallel sublattices and rotates their spin under time reversal. In conventional AFs this sublattice interchanging symmetry involves parity (inversion) or translation combined with time reversal, which leads to a non-relativistic spin-degenerate band structure through

[1]Institut für Physik, Johannes Gutenberg-Universität Mainz, 55099 Mainz, Germany. [2]Inst. of Physics Academy of Sciences of the Czech Republic, Cukrovarnická 10, Praha 6, Czech Republic. [3]Paul Scherrer Institut, CH-5232 Villigen PSI, Switzerland. [4]Ernst Ruska-Centre for Microscopy and Spectroscopy with Electrons, Forschungszentrum Jülich, 52425 Jülich, Germany. [5]Department of Physics and Astronomy, George Mason University, Fairfax, VA 22030, USA. [6]Center for Quantum Science and Engineering, George Mason University, Fairfax, VA 22030, USA. [7]Centre for Quantum Spintronics, Norwegian University of Science and Technology NTNU, 7491 Trondheim, Norway. [8]Department of Physics, Texas A&M University, College Station, TX 77843-4242, USA. ✉e-mail: jourdan@uni-mainz.de

Kramer's theorem. In contrast, in AMs the crystallographic sites of these anti-aligned sublattices are only interconnected through a rotation (proper or improper, symmorphic or nonsymmorphic), enforcing the alternating spin-splitting of exchange origin in the band structure, whose non-magnetic phase is often dominated by crystal fields (CEFs). However, due to symmetry reasons, the band splitting appears only along low-symmetry paths within the Brillouin zone. Comparable effects have been theoretically predicted for materials exhibiting non-collinear magnetic ordering, which breaks both time and crystal symmetry, as discussed in previous works[11–16].

The spin-splitting phenomenon observed in AMs holds profound relevance in the context of spintronics, as it has the potential to generate substantial spin polarized currents with long coherence length scales. These currents have been proposed for applications such as manipulating the magnetic order within a ferromagnetic layer positioned on top of an AM layer, the so-called spin splitter effects predicted in Ref. 17. This unique transport effect of AMs has been substantiated through the detection of a spin torque stemming from the anticipated AM material $RuO_2$[18–20]. Additionally, as predicted for AMs[3], the observation of an anomalous Hall effect has been documented not only in $RuO_2$, as discussed in ref. 21, but also in $Mn_5Si_3$ and MnTe[22,23]. For $RuO_2$, recent experimental observation of a magnetic circular dichroism in angle-resolved photoemission spectra have provided evidence for an altermagnetic band structure[24]. However, the magnitude of the altermagnetic band-splitting and the precise positioning of the relevant states with respect to the Fermi surface remain uncharted territories in experimental observations. So far only for MnTe, experimental evidence for altermagnetic band splitting were reported[25–27]. In the context of AMs for spintronics, the primary goal is to attain a substantial spin-splitting within the conduction bands in close proximity to the Fermi surface. While in conventional AFs spin-splitting originates from spin-orbit coupling with maximum observed values of $\simeq 100$ meV[28], for AMs, density functional theory (DFT) calculations have predicted a one order of magnitude larger spin-splitting of the valence bands.

Given the importance of this parameter in the realm of spintronic applications, the validation of this spin-splitting magnitude through experimental means becomes paramount in comprehensively evaluating the potential of altermagnets for spintronics. In this study, we explore the electronic bands in epitaxial thin films of the altermagnet CrSb. We employ spin-integrated soft X-ray angular-resolved photoelectron spectroscopy (SX-ARPES) to probe these bands. Though we do not probe the spin, the observation of band dispersions, which are in agreement with band structure calculations implying spin splitting, provides strong evidence for an altermagnetic band structure of CrSb. Through a direct comparison with band structure calculations in the ordered and the unordered phase, we demonstrate a large magnitude of the spin-splitting of approximately 0.6 eV near the Fermi energy.

## Results and discussion
### Altermagnetic band spin-splitting in CrSb
Among metallic altermagnets, CrSb distinguishes itself by large predicted band splitting[2] and an ordering temperature significantly exceeding room temperature. Hexagonal CrSb orders magnetically at $T_N = 700$ K[29] with ferromagnetic (001) planes, which are coupled anti-ferromagnetically along the easy c-axis[30]. Figure 1, panel **a**, depicts the crystal structure of the compound. Panel **b** zooms into the local crystallographic environment of the magnetic sublattices. Along the c-axis, the triangular arrangements of the Sb atoms above and below each Cr sublattice are rotated by 60° with respect to each other. This is the origin of the for each Cr sublattice different orientation of the anisotropic crystal electric field (CEF). Interchanging the local environment of the sublattices results in a swapping of the spin polarization of the altermagnetically split bands (within a single magnetic domain). Our epitaxial thin films represent an experimental realization of this

crystal structure with an unknown size of sample regions with identical local sublattice environment (see Supplementary Material).

Panel **c** shows the Brillouin zone of CrSb, indicating three paths discussed below.

Previous calculations have predicted significant altermagnetic band splitting, along the low-symmetry path Γ-L[1]. Here, we consider in particular the likewise low symmetry Q-P path, as this direction is best accessible with our experimental geometry. The Q point is situated halfway between the Γ-point and the M-point, the P-point is halfway between the A- and the L-point. Thus, the Q-P path cuts the Γ-L path at its center, where the largest altermagnetic band splitting is expected. The corresponding band structure along P-Q-P, calculated without spin-orbit coupling (SOC), shows a large altermagnetic band splitting with the spin polarization of the bands changing sign at the Q-point (Fig. 1d). SOC induces minor additional band shifts only (panel **e**), thus, it is not significant for the formation of the altermagnetic band splitting indicated by the pink arrows. To demonstrate the role of the exchange interaction for the formation of the altermagnetic band structure, we show in panel **f** a non-magnetic calculation of CrSb. Again, the anisotropic CEF results in an energy splitting of the projections of the electronic states on the crystallographically distinct Cr sublattices. However, compared to the altermagnetic case, the bands show a qualitatively very different dispersion. Thus, the exchange interaction does not just add rigid spin dependent energy shifts to the bands, i. e., it is **k**-dependent. This means that even without an analysis of the spin orientation the altermagnetic state can be clearly distinguished from the non-magnetic state based on the k-dependence of the electronic bands.

The electronic states within high-symmetry planes, which include the Q and P points, display degeneracy. Consequently, along high-symmetry *k*-space paths on these planes, which are conventionally investigated by ARPES, the altermagnetic band splitting is absent. Along these spin degenerate paths, our calculations align closely with previously reported findings[31]. In contrast, for the low-symmetry path that links Q and P, as previously discussed (Fig. 1d, e), substantial spin splitting of the bands is anticipated. To confirm the validity of the band structure calculations, and in particular to probe the magnitude of the altermagnetic band splitting, we perform SX-ARPES investigations of epitaxial CrSb(100) thin films.

### Band structure investigation by SX-ARPES
The analysis of ARPES data hinges on the accurate identification of distinctive directions in reciprocal space. Thus, it becomes imperative to experimentally ascertain the characteristic high-symmetry points within the Brillouin zone. We identify the center of the Brillouin zone (Γ-point) by scanning the photon energy, which corresponds to a scan in *k*-space along the direction perpendicular to the CrSb(100) sample surface (parallel to the Γ-M direction, Fig. 1c). The resulting ARPES intensity $I(\mathbf{k})$ at the Fermi energy is shown in Fig. 2, panel **a**, and correspond to a cut through the Fermi surface in the Γ-M-K plane. A cut through the perpendicular Γ-K-A plane, measured with a fixed photon energy (775 eV) selecting one of the Γ-points from panel **a**, is shown in panel **b**. Both 2-dimensional constant-energy $I(\mathbf{k})$ cuts at the Fermi energy are in good agreement with the calculated 3-dimensional Fermi surface shown in panel **c**, providing evidence for the validity of the calculation and quality of the samples. The observed strong variation of the ARPES intensity from different Brillouin zones will be discussed below.

We now discuss specific directions in *k*-space with and without altermagnetic band splitting and contrast the ARPES intensity $I(E, \mathbf{k})$ with band structure calculations for selected paths traversing the Brillouin zone. To enhance the comparability with the experimental data, we incorporate spin-orbit coupling (SOC) into the calculations. While the angle-dependent photoemission intensity does not precisely replicate the electronic band structure owing to distinct

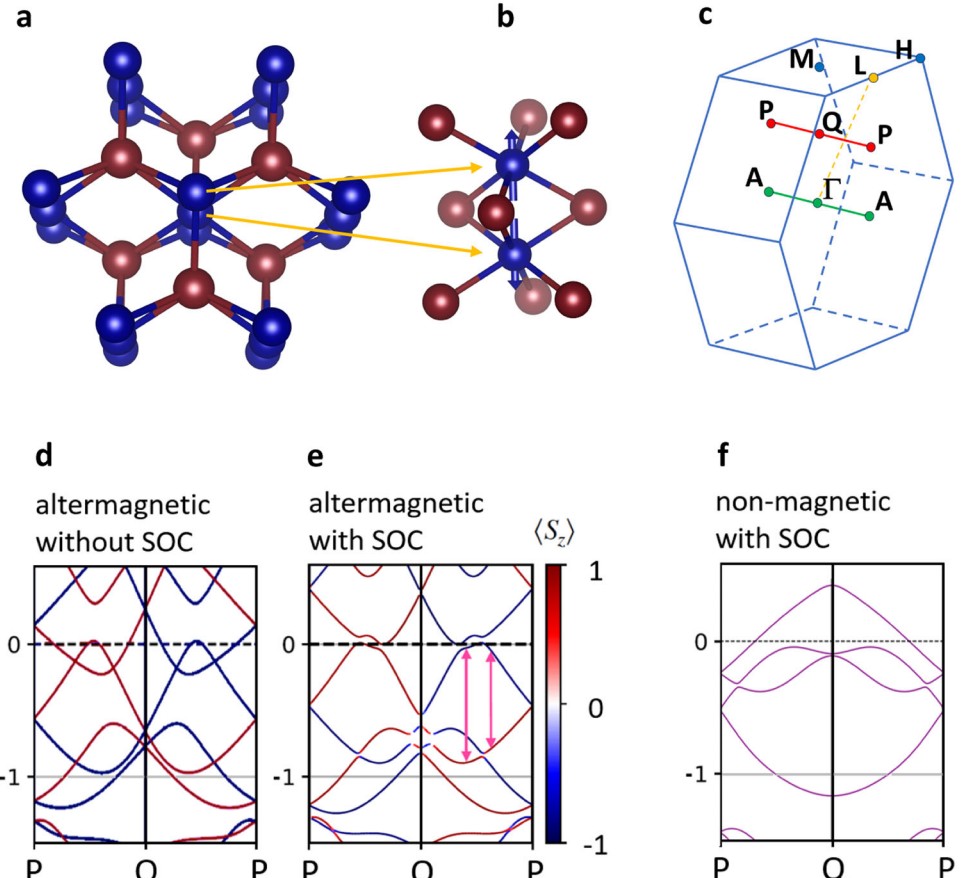

**Fig. 1 | Structure of CrSb. a** Crystal structure. Blue: Cr atoms, Red: Sb atoms. Drawn with VESTA[42]. **b** Local environment of the Cr sublattices with antiparallel alignment of the magnetic moments. **c** Brillouin zone of CrSb showing, e.g., the P-Q-P path, for which altermagnetically split bands are expected. **d** Altermagnetic band structure calculations without SOC for the P-Q-P path. The color of the bands indicates the spin polarization. **e** Altermagnetic band structure calculations with SOC for the P-Q-P path. **f** Band structure of the non-magnetic state (with SOC).

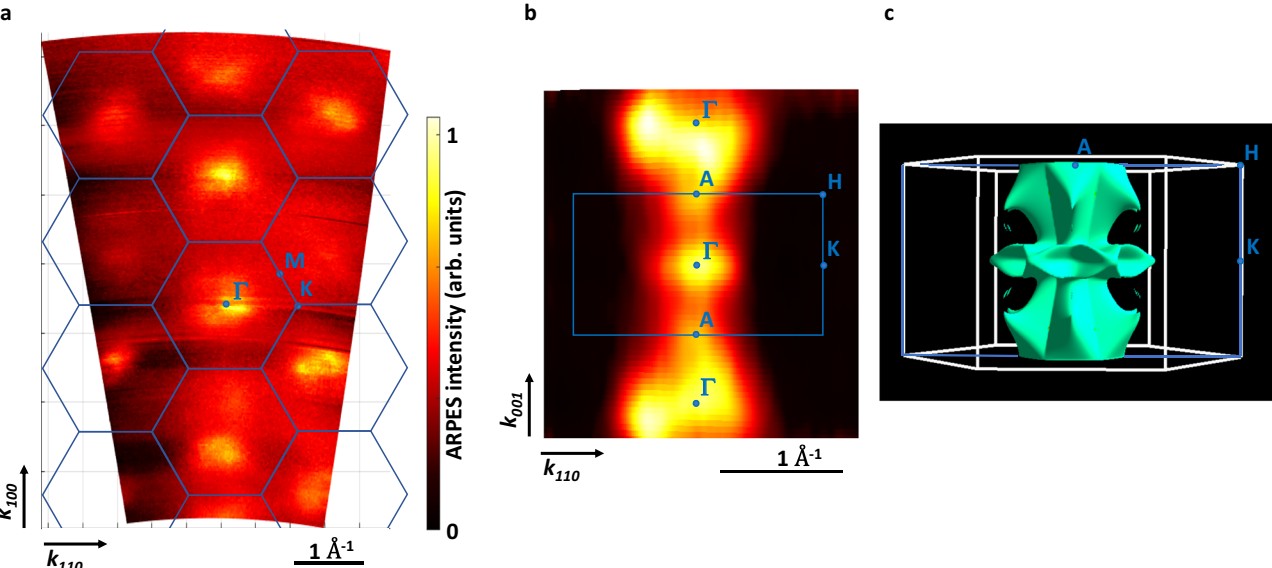

**Fig. 2 | Experimental and theoretical Fermi surface of CrSb.** Comparison of the SX-ARPES intensity $I(k)$ at the Fermi energy obtained in the Γ-M-K plane (panel **a**) and in the Γ-K-A plane (panel **b**) with the calculated 3-dimensional Fermi surface shown in panel **c**. The corresponding Brillouin zone and high symmetry points are superimposed for clarity.

photoemission matrix elements[32], it does allow for the direct observation of distinctive band structure features. To investigate the altermagnetic band splitting in the CrSb(100) films using ARPES, we specifically opt for the Q-P path, as shown in Fig. 1. This chosen path is aligned parallel to the sample surface, facilitating the direct imaging of

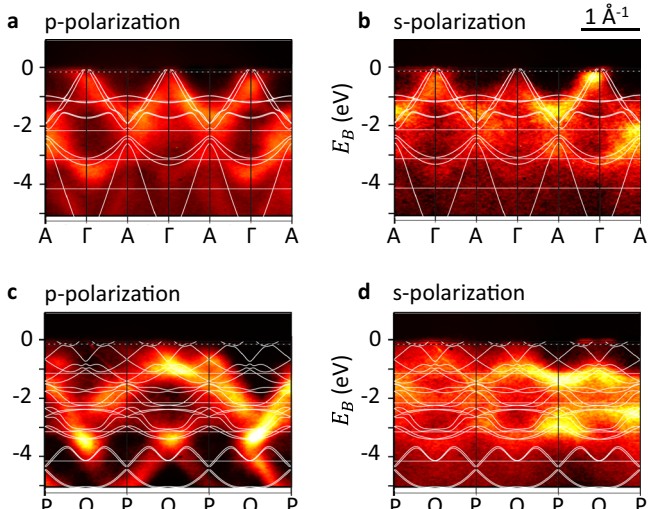

**Fig. 3 | Wide energy range ARPES intensity and band structure calculations.** The SX-ARPES intensity $I(E, k)$ is shown with the corresponding superimposed band structure calculations. **a** High symmetry Γ-A path with p-polarized photons. **b** Γ-A path with s-polarized photons. **c** Low-symmetry Q-P path with p-polarized photons. **d** Q-P path with s-polarized photons.

the corresponding ARPES intensity $I(E, \mathbf{k})$ with our experimental detector setup. This low-symmetry path with large altermagnetic spin-splitting is parallel to the high symmetry Γ-A path, which, by symmetry, shows no spin splitting. Consequently, we can measure both paths in $k$-space with the same experimental geometry but different photon energies, selecting different values of $k_z$. Since the wave vector $\mathbf{k}$ of the photoemitted electrons resides within a crystallographic mirror plane of CrSb (as illustrated in Supplementary Fig. 5b), we take into consideration the parity selection rules[33] by acquiring spectra with both p- and s-polarized photons. Furthermore, the ARPES data contains an $\pm k$ asymmetry induced by the photon momentum due to the 9° grazing photon incidence from the positive $k$-value side in Figs. 3 and 4 (see the experimental geometry in Supplementary Fig. 5a).

Figure 3 presents a comparison between the ARPES intensity $I(E, \mathbf{k})$ and band structure calculations for both $k$-space paths discussed above.

The background subtracted ARPES raw data covering three Brillouin zones is shown together with the superimposed band structure calculations in the same plot. To provide a comprehensive understanding of the relationship between ARPES intensity and band structure calculations for CrSb, we first discuss the overall data trends acquired over a wider energy range.

For the high-symmetry Γ-A path (panels **a** and **b**), the consistency between the ARPES data and the band structure calculations is evident, irrespective of the chosen Brillouin zone and photon polarization. In all figures, the calculated band structure's energy scale has been rigidly shifted by ≈150 meV for better energy alignment.

For the low-symmetry Q-P path (panels **c** and **d**), the situation is more complex. Here, a pronounced polarization dependence, which is discussed in the Supplementary Information in the framework of selection rules, and a forward-backward scattering asymmetry

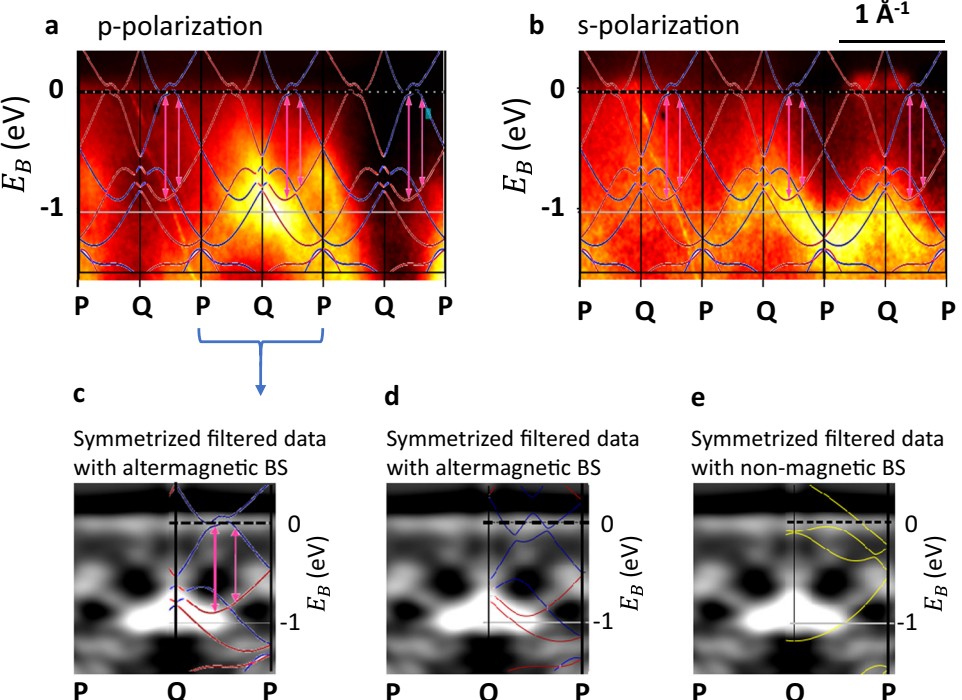

**Fig. 4 | Band splitting near the Fermi energy.** The spin integrated SX-ARPES intensity $I(E, \mathbf{k})$ just below the Fermi energy is shown with the corresponding superimposed spin resolved band structure calculations. **a**: High symmetry P-Q path with p-polarized photons. **b**: P-Q path with s-polarized photons. Panels **c** and **d** show the symmetrized ARPES intensity (after application of an Laplacian filter enhancing the visibility of the bands) from the central Brillouin zone for

p-polarization with the superimposed altermagnetic band structure calculation. **c** compares with a calculation based on bulk lattice parameters[34], **d** with a calculation based on the experimentally determined parameters of the CrSb thin films, which results in improved agreement. Panel **e** show the same ARPES data superimposed with a non-magnetic band structure calculation. Here, the missing agreement emphasises the importance of the exchange interaction.

inherent to the ARPES geometry (see Methods) is discernible. Furthermore, for this path, a strong alternation of the ARPES intensity from Brillouin zones to Brillouin zone is observed. These effects are embedded in the matrix elements describing the transition from the occupied to the photoemitted electronic states[32]. Nevertheless, it is possible to provide an intuitive explanation: the alternating ARPES intensity likely arises from an interference effect associated with the periodic arrangement of the Cr atoms within the CrSb unit cell (as depicted in Fig. 1a). These atoms alone correspond to a Brillouin zone that is doubled along the c-direction compared to the actual full CrSb cell. The inclusion of the Sb atoms can be viewed as a minor perturbation, which primarily preserves the physical scattering potential of the Cr-only states. However, formally, it leads to the folding of the Cr bands into a Brillouin zone half the size of the original CrSb Brillouin zone. Consistently, our calculations shown in the Supplementary Information demonstrate a strong Cr d-orbital character of the electronic states. In ARPES, this translates into a periodicity of the photoemission intensity that aligns with the doubled formal Brillouin zone. Thus, bands with a strongly alternating ARPES intensity have mainly Cr character. However, when taking into account both photon polarizations and all Brillouin zones collectively, the experimental data is consistent with the band structure calculations across a broad spectrum of binding energies for the Q-P path.

## Alternagnetic band splitting observed by SX-ARPES

Next, we focus our attention to the energy range in proximity to the Fermi energy, a region where the alternagnetic band splitting is anticipated and holds particular relevance for practical applications. Within our experimental setup, direct observation of the alternagnetic band splitting becomes feasible for the Q-P path, as depicted in Fig. 4.

The most significant splitting is highlighted by the pink arrows in the superimposed band structure calculation. Within the ARPES data, the majority of bands are most distinctly observable within the central Brillouin zone, particularly when employing p-polarized photons, as shown in panel a and backward scattering (photon grazing incidence from positive k-value direction). In this context, the energetically lower branch of alternagnetically split band is prominently discernible in the experimental data. Conversely, when employing s-polarized photons, as displayed in panel b, this lower branch is notably absent, indicating the presence of an essentially even-parity state with respect to the scattering plane as discussed in detail in the Supplementary Information. In Supplementary Fig. 5, we show that the alternagnetically split band has mainly the character of a Cr $d_{x^2-y^2}$ orbital. The anticipated energetically higher branch of the split band is less conspicuous in the experimental data and becomes overshadowed by the forward-backward scattering asymmetry. Therefore, in panels c to e, we present the ARPES results from the central Brillouin zone acquired with p-polarized photons after symmetrization and filtering. With this procedure, both branches of the alternagnetically split band become clearly discernible. To enhance visibility and to pinpoint the centers of the split band branches with greater precision, we have computed a Laplacian image the ARPES raw data. The application of the Laplace operator enhances the positions of maximum curvature within the ARPES intensity (see Methods). Consequently, the distinctive features of both branches become pronounced, allowing for a comprehensive comparison with the band structure calculations superimposed onto the experimental data.

In panel c, the filtered ARPES intensity is partially superimposed with the alternagnetic band structure calculation for comparison, showing a good agreement of the main features. In particular, the alternagnetic band splitting, as indicated by the pink arrows in the calculation, is now conspicuously evident in the experimental data. The lower branch of this splitting, which was already discernible in the non-symmetrized ARPES raw data (central Q-P path in Fig. 4a) aligns remarkably well with a region of elevated ARPES intensity. Conversely, the upper branch appears faint in the raw data, but becomes now distinctly discernible. Based on the experimental results, we estimate the alternagnetic band splitting of our epitaxial CrSb(100) thin films to amount to 0.6 eV.

The experimentally identified upper branch is positioned approximately 200 meV lower in energy compared to the band structure calculated assuming the published hexagonal lattice parameters ($a = 4.123$ Å, $c = 5.47$ Å[34]). However, our thin films feature slightly different lattice parameters of $a = 4.1$ Å and $c = 5.58$ Å (see Supplementary Information). Panel d shows that with these parameters the calculated upper branch of the alternagnetically split band is pushed towards the lower branch resulting in an improved agreement with the experimental data. With such a strong effect of strain on the upper branch, the remaining discrepancies between theory and experiment could be explained by additional small lattice distortions in the surface region of the CrSb thin film mainly probed by SX-ARPES (top 1-2 nm).

In the Supplementary Information, we additionally show the symmetrized ARPES intensity (raw data) before application of the Laplacian filter. In such a representation, the upper band is barely noticeable. Nevertheless, the magnitude of the alternagnetic band splitting can also be estimated directly from this raw data and amounts consistently to 0.6 eV.

For comparison, we show in panel d the superposition of the non-magnetic band structure calculation (see also Fig. 1f) with the experimental ARPES data, showing a lack of consistency as expected for an alternagnetic material. This is most obvious in the region of highest ARPES intensity around the Q point at a binding energy of $\simeq 1$ eV and in the region where the upper branch of the alternagnetically split band has been identified in panel c.

In summary, we have combined experimental SX-ARPES investigations with first-principles calculations to identify the alternagnetic band structure of CrSb. Good agreement was obtained, both regarding the course of the dispersion branches as well as the polarization dependence of the ARPES intensity. In particular, we have detected band degeneracy at specific k-space points situated on high-symmetry planes perpendicular to the c-axis. These planes encompass Q within the central plane containing the Γ point and P, which is positioned at the boundary of the Brillouin zone. Along the trajectory that connects these points, we observe the anticipated alternagnetic band splitting.

The fundamental hallmark of alternagnetism lies in such spin splitting of electronic valence bands, for which our results furnish direct experimental evidence focusing on CrSb. While similar results on the band structure were reported very recently for MnTe[26], CrSb stands out due to the significant magnitude of the band splitting, quantified to be 0.6 eV in our measurements, and the notable energetic placement of the strongly split band positioned just below the Fermi energy. This specific energetic positioning holds crucial significance in the context of potential spin-polarized currents. Consequently, our results establish the fundamental groundwork for potential spintronic applications leveraging the emerging class of alternagnetic materials.

# Methods
## Film preparation
Epitaxial CrSb(100) thin films of 30 nm thickness were grown by dc magnetron sputtering from a single multi-segment Cr/Sb target on GaAs(110) substrates. After deposition at a substrate temperature of $\simeq 300$ °C, the sample was annealed in-situ at a temperature of $\simeq 400$ °C for 15 min. As shown in the Supplementary Information by X-ray and electron diffraction as well as by transmission electron microscopy, the thin films are fully epitaxial with the in-plane hexagonal c-axis of CrSb(100) aligned parallel to the in-plane (001)-direction of the GaAs(110) substrate.

## SX-ARPES measurements and data evaluation

After the electron diffraction (reflection high-energy electron diffraction, RHEED) based confirmation of a structurally well-ordered sample surface (see Supplementary Information), the epitaxial thin films were transported to the ARPES endstation using a vacuum suitcase. The main advantage of SX-ARPES regarding investigations of the three-dimensional band structure of CrSb is the high intrinsic resolution in the surface-perpendicular electron momentum[35]. The SX-ARPES investigations were performed at the ADRESS beamline[36,37] of the Swiss Light Source. The ARPES endstation uses an experimental geometry with 9° grazing light incidence angle. As the direction of light incidence and the angle dispersive direction (detector slit orientation) are in the same plane (see Supplementary Fig. 5a), the photon momentum creates a forward-backward scattering asymmetry in the photoemission intensity. The measurements were conducted at a temperature of 12 K, with varying the photon energy from 320 eV to 1000 eV. Γ points were identified, e.g., at photon energies of 775 eV and 935 eV. The total energy resolution including thermal broadening varied from 50 meV to 100 meV in this energy range. All ARPES data shown in this manuscript was visualized using the Matlab program ARPESView (by Strocov (https://www.psi.ch/en/sls/adress/manuals)), which subtracts an angle-integrated background from the raw data. Beyond this, panel **c** of Fig. 4 shows symmetrized data, i. e., the ARPES image was mirrored along the $k_z = 0$ axis and added to the original data. The band enhancement shown in the right figure of panel **c** was obtained using the Fiji distribution of the image processing software ImageJ, specifically by applying the function FeatureJ: Laplacian.

## Band structure calculations

The experimental results were compared with calculations based on ab initio spin-density functional theory with local-density approximation. We performed the ab initio electronic structure calculations for CrSb (Space group $P6_3/mmc$, No. 194) with the pseudopotential Vienna Ab initio Simulation Package (VASP)[38], using the Perdew–Burke–Ernzerhof (PBE) + SOC generalized gradient approximation (GGA)[39-41]. For the order phase, we used a $6 \times 6 \times 5$ k-grid, an energy cut-off of 400 eV, switched off symmetrization and converged the self-consistency and energy band calculations within an energy convergence criterion of $10^{-5}$ eV. For the nonmagnetic phase, we used a $15 \times 15 \times 9$ k-point grid, an energy cut-off of 400 eV, and a convergence criterion of $10^{-6}$ eV. The lattice parameters are $a = b = 4.103$ Å and $c = 5.463$ Å, corresponding to experimental values.

## Data availability

The raw data of the XRD and ARPES line graphs shown in this study have been deposited in the Zenodo database under accession code 10.5281/zenodo.10531915.

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

## Acknowledgements

We acknowledge funding by the Deutsche Forschungsgemeinschaft (DFG, German Research Foundation) - TRR 173 - 268565370 (projects A05 (M.J.), with contribution from A01 (M.K.)), by the Horizon 2020 Framework Program of the European Commission under FET-Open Grant No. 863155 (s-Nebula) (M.K., M.J.), by EU HORIZON-CL4-2021-DIGITAL-EMERGING-01-14 programme under grant agreement No. 101070287 (SWAN-on-chip) (M.K., M.J.), and by the TopDyn Center (M.K.). We acknowledge the Swiss Light Source for time on beamline ADRESS under Proposal 20222058 (M.J.). L.S and A.B.H. acknowledges support from the Johannes Gutenberg-Universität Mainz TopDyn initiative. R.J.U., W.H.C. and V.K.B. acknowledge funding from the Deutsche Forschungsgemeinschaft (DFG) grant no. TRR 173 268565370 (project A03). A.B.H., A.C. and J.S.acknowledge funding from Deutsche Forschungsgemeinschaft (DFG) grant no. TRR 288 - 422213477 (Projects A09 and B05). The STEM investigations were funded by the European Union's Horizon 2020 Research and Innovation Programme under grant agreement 856538 (project "3D MAGIC") (M.K., R.E.D.-B.). We acknowledge support by H.-J. Elmers (JGU Mainz) for the Laplacian filter application.

## Author contributions

S.R. and M.J. wrote the paper with contributions by J.S.; S.R., L.O. and M.J. prepared the samples and performed and evaluated the SX-ARPES investigations supported by V.N.S. and P.C.; L.S., A.B.H., R.J.U., W.H.C., V.K.B. and A.C. provided the band structure calculations; S.D. provided the Fermi surface calculation; T.D., W.S. and R.E.D-B contributed the STEM investigations. M.K. contributed to the discussion of the results and provided input; M.J. coordinated the project.

## Funding

## Competing interests

The authors declare no competing interests.
