## [Peer Review File · Nature Communications]

Direct observation of altermagnetic band splitting in CrSb thin filmsREVIEWER COMMENTS

Reviewer #1 (Remarks to the Author):

The article provides an interesting and detailed characterization of the CrSb thin films. However, I have some criticism that should be addressed by the authors before publication: As a summary, the article contains valuable information about these CrSb thin films. Especially both experiments and theories. However, in the discussion of basic electronic properties, the author points that "Additional shifts of specific bands may be attributed to electronic correlation effects not accounted for in the calculation." They should also convince me that the electronic correlation effects are not achievable in experimentally, otherwise the electronic correlation effects should be accounted for in the calculation. After changes according to my comments have been made accordingly, otherwise, I might not recommend publication in Nature Communications.

- 1, The physical properties are closely related to their structure, and the symmetry of the CrSb thin films should be analyzed.
- 2, They consider in particular the likewise low symmetry Q-P path, as this direction is best accessible with our experimental geometry. Can this low symmetry path fully describe the intrinsic properties of materials?
- 3, Why selecting different values of k_z , when they measure both paths in k-space?
- 4, The inclusions that of the Sb atoms can be viewed as a minor perturbation, which primarily preserves the physical scattering potential of the Cr-only states should be given in the theories.
- 5, The theories show in panel d the superposition of the non-magnetic band structure calculation (see also Fig.1f) with the experimental ARPES data, showing a lack of consistency. Why? Their explanations are not convincing me. Theoretical and experimental inconsistencies require special attention and declaration.

Reviewer #2 (Remarks to the Author):

The authors reported the first observation of the spin splitting (originated by non-relativistic spin splitting) in CrSb bulk using ARPES. My opinion of the paper is quite positive, but I would like some moderate revisions before acceptance.

Below, you find my questions/remarks:

- 1) it seems that the low-symmetry path P-Q-P is halfway between the gamma point and the border of the Brillouin zone. However, this is not explicitly written in the paper, please confirm it and add it. The authors should also find a way to highlight that the QP and PQ paths are inequivalent.
- 2)The authors wrote that the nonrelativistic spin splitting is 0.6 eV, however, it seems to me that the maximum spin-splitting is even larger in the presented figures. Please make the text uniform with the figures of this paper.
- 3) What is the effect of SOC on the non-relativistic spin splitting? Do we have a slight increase in the gap?
- 4) was the anomalous Hall effect in CrSb ever reported by somebody?

Reviewer #3 (Remarks to the Author):

The authors aim to elucidate the existence of non-relativistic spin-splitting in an antiferromagnetic CrSb crystal. Non-relativistic spin-splitting holds promise for future spintronic applications compared to relativistic counterparts, whose sizes may be restricted to be small due to the typically tiny spin-orbit coupling strength, especially when transition metal atoms are included in the crystals. Investigating the spin-split band structures of altermagnets is crucial.

Despite the authors' optimistic claims that non-relativistic spin-splitting was observed by soft-X-ray ARPES, I am skeptical about their claims. The key result leading to the conclusion of this work is presented in Fig. 4, showing p- and s-polarized radiation ARPES. The data statistics are insufficient to support their key conclusion. There is no experimental counterpart for the predicted upper branch of the spin-split band. The application of Laplacian filtering does not result in visualizing the upper branch, generating a discrete spotty distribution as shown in Fig. 4c.

It is recommended to add line profiles of energy and moment distributions to show intensity peaks.

The reasons for the ambiguous data might be the inherent low momentum and energy resolutions of soft-X-ray ARPES and contaminants deposited during sample transportation using the suitcase. It is unclear whether the sample was annealed after installation into the ARPES chamber.

Spin-resolved band dispersions are necessary for the direct observation of spin-splitting bands of the altermagnet.

The discussion about orbital characters in splitting bands are missing though the authors measured both p- and s-polarized cases.

Soft x-ray ARPES can often be utilized, and band-dispersion results are usually plotted with the calculations. A similar observation was reported for RuO₂ using the MCD-ARPES technique. <https://arxiv.org/abs/2306.02170>

Based on the above concerns, I cannot recommend the current manuscript for publication in Nature Communications.

We would like to thank the reviewers for their insightful comments regarding our manuscript, which helped us to improve it considerably.

Please find below our point-by-point response to all reviewer's concerns (8 pages). For improved readability, citations from the reviewer reports are printed in black, our reply is printed in grey, and new text added to the manuscript is printed in blue (here and in the manuscript).

The following new figures were added to the manuscript (please see there)

Main text: Fig. 4, new panel d (old panel d is now panel e)

Supplementary Information: Figs. S1, panels b and c; S3; S4 (old S3 is now S5).

Reviewer #1 (Remarks to the Author):

The article provides an interesting and detailed characterization of the CrSb thin films. However, I have some criticism that should be addressed by the authors before publication:

As a summary, the article contains valuable information about these CrSb thin films. Especially both experiments and theories. However, in the discussion of basic electronic properties, the author points that "Additional shifts of specific bands may be attributed to electronic correlation effects not accounted for in the calculation." They should also convince me that the electronic correlation effects are not achievable in experimentally, otherwise the electronic correlation effects should be accounted for in the calculation.

Within density functional theory (DFT), correlation effects can partially be accounted for by the introduction of an adjustable parameter (LDA+U), which is related to the Coulomb interaction in the Hubbard model. We now have done such a calculation assuming $U = 3$ eV, which resulted in an increase of the band splitting (see figure below), which is not in agreement with our experimental results. So correlation effects are not of major importance for the band structure of CrSb.

Based on this new insight, we have removed the phrase mentioning possible correlation effects from the manuscript (p. 5).

In search for other explanations for the in the LDA calculation larger antiferromagnetic band splitting compared to the experimental results, we now discuss the effect of epitaxial strain on the band structure. We have performed an additional LDA calculation using the experimentally determined lattice parameters of our epitaxial CrSb thin films instead of the literature values. As discussed below referring to your remark (1), the a-axis is shortened and the c-axis elongated compared to the previously published lattice parameters of bulk CrSb. Our LDA calculation with these real lattice parameters resulted in a clearly improved agreement of the theoretical and experimental data including in particular the magnitude of the antiferromagnetic band splitting. We added the **new panel d to Fig. 4** of the manuscript and added the following text on page 5:

“The experimentally identified upper branch is positioned approximately 200 meV lower in energy compared to the band structure calculated assuming the published hexagonal lattice parameters ($a = 4.123 \text{ \AA}$, $c = 5.47 \text{ \AA}$ [24]). However, our thin films feature slightly different lattice parameters of $a = 4.1 \text{ \AA}$ and $c = 5.58 \text{ \AA}$ (see Supplementary Information). Panel d shows that with these parameters the calculated upper branch of the antiferromagnetically split band is pushed towards the lower branch resulting in an improved agreement with the experimental data. With such a strong effect of strain on the upper branch, the remaining discrepancies between theory and experiment could be explained by additional small lattice distortions in the surface region of the CrSb thin film mainly probed by SX-ARPES (top 1-2 nm), ~~presumably due to the effect of electronic correlations not covered in the calculation. Thus, we estimate the antiferromagnetic band splitting of our epitaxial CrSb(100) thin films to amount to 0.6 eV.~~”

After changes according to my comments have been made accordingly, otherwise, I might not recommend publication in Nature Communications.

1, The physical properties are closely related to their structure, and the symmetry of the CrSb thin films should be analyzed.

We added on page 2 of the revised manuscript:

“This is the origin of the for each Cr sublattice different orientation of the anisotropic crystal electric field (CEF). Interchanging the local environment of the sublattices results in a swapping of the spin polarization of the antiferromagnetically split bands (within a single magnetic domain). Our epitaxial thin films represent an experimental realization of this crystal structure with an unknown size of sample regions with identical local sublattice environment (see Supplementary Information).

Correspondingly, we **added the panels b and c to Fig. S1** of the Supplementary Information. These additional x-ray diffraction results provide further evidence that our samples are indeed grown with the crystal structure shown in Fig. 1 of the main manuscript. We added to the text of the Supplementary Information: “All observed peaks, both in specular (panel a) as well as in off-specular geometry (panel b) agree well with Powder-Cell XRD simulations assuming a hexagonal CrSb crystal structure with space group no. 194 and lattice parameters $a = 4.09 \text{ \AA}$ and $c = 5.65 \text{ \AA}$, as shown in Fig. 1. The XRD rocking curve shown in panel c indicates a moderate mosaicity (tilt range of the 001-axis) of $\approx 0.5^\circ$ resulting in broadening of the ARPES data in k_{\parallel} .”

Furthermore, we added a discussion of the local crystallographic environment of the Cr sublattices of CrSb to the Supplementary Information:

“It is important to acknowledge that due to the inherent thickness averaging within the TEM lamella, the interchangeability of the local Sb environment within the antiferromagnetic Cr sublattices remains beyond visualization. **Thus, we cannot make any statement about the size of sample regions with identical local Sb environment of the Cr sublattices.**”

2, They consider in particular the likewise low symmetry Q-P path, as this direction is best accessible with our experimental geometry. Can this low symmetry path fully describe the intrinsic properties of materials?

The antiferromagnetic band splitting is in general absent in high symmetry planes of the Brillouin zone (BZ) (Smejkal et al., Phys. Rev. X, 12, 031042 (2022)). E.g., all BZ boundaries can be equivalently described by k-vectors originating from adjacent Gamma points in the repeated zone scheme. However, due to the different directions of those k-vectors they would in general be associated with

different antiferromagnetic spin splitting. Thus, the splitting needs to be zero for those planes in k-space. To emphasize this issue, we added the following statement to the manuscript (p. 1):

However, due to symmetry reasons, the band splitting appears only along low-symmetry paths within the Brillouin zone.

Correspondingly, the Q-P path we focused on is the best accessible, which also contains the largest expected antiferromagnetic band splitting. This large band splitting was theoretically identified on Ref. [3] for the Gamma-L path shown in Fig. 1c. Our Q-P path cuts this Gamma-L path exactly at the position of the largest splitting.

3, Why selecting different values of k_z , when they measure both paths in k-space?

In angular resolved photoemission spectroscopy, k parallel to the sample surface is conserved based on translational invariance. However, k perpendicular to the sample surface is not conserved, as the transition from the sample to vacuum breaks the symmetry. Nevertheless, by varying the photon energy, k_z (perpendicular to the surface) can be varied. This is best understood within the 3-step model of photoemission, in which a transition from an occupied state to an unoccupied free-electron like state far above the Fermi energy is considered.

Our CrSb samples grow in (100)-orientation, i.e. the BZ is oriented on the substrate as shown in Fig. 1c. In this case k_z corresponds to the Gamma-M direction. So by selecting different photon energies, any path parallel Gamma-A can be moved into the angular dispersive k_{parallel} direction of the spectrometer, as the photon energy determines the k_z value.

To make this more clear in the manuscript, we added the following statement (p. 3):

“We identify the centre of the Brillouin zone (Γ -point) by scanning the photon energy, which corresponds to a scan in k-space along the direction perpendicular to the CrSb(100) sample surface (parallel to the Gamma-M direction, see Fig. 1c)”.

4, The inclusions that of the Sb atoms can be viewed as a minor perturbation, which primarily preserves the physical scattering potential of the Cr-only states should be given in the theories.

We have now added the **new Figure S4** and corresponding text to the Supplementary Information, which shows the strong Cr-character of the electronic states based on band structure calculations. In the main text (p. 4), we added:

“However, formally, it leads to the folding of the Cr bands into a Brillouin zone half the size of the original CrSb Brillouin zone. Consistently, our calculations shown in the Supplementary Information demonstrate a strong Cr d-orbital character of the electronic states.”

Furthermore, we added to the Supplementary Information:

“**Orbital character of the valence states** - To identify the orbital character of the electronic bands showing the antiferromagnetic splitting, we show in Fig. S4a the projections on the Cr as well as on the Sb atomic orbitals. The dominant Cr character is obvious from the comparison of the calculated spectral weights of the orbitals. This result is fully consistent with the experimentally observed periodicity of the ARPES intensity from BZ to BZ, as discussed in the main text. Furthermore, we show in panel b the projections of the electronic bands on the s-, p-, and d-orbitals of Cr. We observe, that the major contribution originates from d-orbitals, which we discuss below in the framework of selection rules in photoemission spectroscopy.”

5, The theories show in panel d the superposition of the non-magnetic band structure calculation (see also Fig.1f) with the experimental ARPES data, showing a lack of consistency. Why? Their

explanations are not convincing me. Theoretical and experimental inconsistencies require special attention and declaration.

We have shown the non-magnetic band structure calculation in panel f to demonstrate the strong effect of the magnetic order on the band structure. As such, it is essential that the experimental ARPES data does NOT fit with the non-magnetic panel f calculation, but only with the altermagnetic (panel e) calculations. This demonstrates that not only the local CEF of the two Cr sublattices determines the band structure, but also the exchange interaction. Please note, that experimental ARPES data can only be obtained below the magnetic ordering temperature of CrSb, which is 700 K [27].

In the revised version, we now improved the understandability of our manuscript in this respect by adding:

“To demonstrate the role of the exchange interaction for the formation of the altermagnetic band structure, we show in panel f a non-magnetic calculation of CrSb. Again, the anisotropic CEF results in an energy splitting of the projections of the electronic states on the crystallographically distinct Cr sublattices. However, compared to the altermagnetic case, the bands show a qualitatively very different dispersion. Thus, the exchange interaction does not just add rigid spin dependent energy shifts to the bands, i. e., it is k-dependent. **This means that even without an analysis of the spin orientation the altermagnetic state can be clearly distinguished from the non-magnetic state based on the k-dependence of the electronic bands.**”

Furthermore, we added

“For comparison, we show in panel d the superposition of the non-magnetic band structure calculation (see also Fig. 1f) with the experimental ARPES data, showing a lack of consistency **as expected for an altermagnetic material.**”

And to the caption of Fig. 4:

“Panel d show the same ARPES data superimposed with a non-magnetic band structure calculation. **Here, the missing agreement emphasises the importance of the exchange interaction.**”

Reviewer #2 (Remarks to the Author):

The authors reported the first observation of the spin splitting (originated by non-relativistic spin splitting) in CrSb bulk using ARPES. My opinion of the paper is quite positive, but I would like some moderate revisions before acceptance.

Below, you find my questions/remarks:

1) it seems that the low-symmetry path P-Q-P is halfway between the gamma point and the border of the Brillouin zone. However, this is not explicitly written in the paper, please confirm it and add it.

Yes, this is correct. We now clarify this issue by adding the following statement to the manuscript (p. 2):

“Here, we consider in particular the likewise low symmetry Q-P path, as this direction is best accessible with our experimental geometry. **The Q point is situated halfway between the Γ -point and the M-point, the P-point is halfway between the A- and the L-point. Thus, the Q-P path cuts the Γ -L path at its center, where the largest altermagnetic band splitting is expected.**”

The authors should also find a way to highlight that the QP and PQ paths are inequivalent.

This is indeed a very special property of altermagnets and we now added to the section referring to Fig. 1 (p. 2):

“The corresponding band structure **along P-Q-P**, calculated without spin-orbit coupling (SOC), shows

a large altermagnetic band splitting with the spin polarization of the bands changing sign at the Q-point (Fig. 1d).”

2)The authors wrote that the nonrelativistic spin splitting is 0.6 eV, however, it seems to me that the maximum spin-splitting is even larger in the presented figures. Please make the text uniform with the figures of this paper.

The spin splitting in the calculations based on the previously published bulk lattice parameters of CrSb is indeed larger, but the upper band appears lower in energy in the experiments. Thus, the experimentally determined band splitting is 0.6 eV:

However, based on a strain analysis of our epitaxial thin films we have now added an additional band structure calculation using the experimentally determined modified lattice parameters of CrSb (please see our reply to reviewer 1). With these parameters the calculated gap approaches the experimental result (see new Figure 4d).

To clarify this, we added to the manuscript (p. 5):

“The experimentally identified upper branch is positioned approximately 200 meV lower in energy compared to the band structure calculated assuming the published hexagonal lattice parameters ($a = 4.123 \text{ \AA}$, $c = 5.47 \text{ \AA}$ [24]). However, our thin films feature slightly different lattice parameters of $a = 4.1 \text{ \AA}$ and $c = 5.58 \text{ \AA}$ (see Supplementary Information). Panel d shows that with these parameters the calculated upper branch of the altermagnetically split band is pushed towards the lower branch resulting in an improved agreement with the experimental data. With such a strong effect of strain on the upper branch, the remaining discrepancies between theory and experiment could be explained by additional small lattice distortions in the surface region of the CrSb thin film mainly probed by SX-ARPES (top 1-2 nm).”

3) What is the effect of SOC on the non-relativistic spin splitting? Do we have a slight increase in the gap?

Comparing Figs. 1d and 1e, the avoided level-crossing with SOC results in a slight decrease of the gap.

4) was the anomalous Hall effect in CrSb ever reported by somebody?

We are not aware of any observation of an AHE in CrSb.

From a symmetry point of view, no AHE is possible in CrSb as long as the magnetic moments are aligned along the easy c-axis of the hexagonal CrSb cell. How the magnetic moments should be tilted out of this easy direction is not obvious.

Reviewer #3 (Remarks to the Author):

The authors aim to elucidate the existence of non-relativistic spin-splitting in an antiferromagnetic CrSb crystal. Non-relativistic spin-splitting holds promise for future spintronic applications compared to relativistic counterparts, whose sizes may be restricted to be small due to the typically tiny spin-orbit coupling strength, especially when transition metal atoms are included in the crystals.

Investigating the spin-split band structures of altermagnets is crucial.

Despite the authors' optimistic claims that non-relativistic spin-splitting was observed by soft-X-ray ARPES, I am skeptical about their claims. The key result leading to the conclusion of this work is presented in Fig. 4, showing p- and s-polarized radiation ARPES. The data statistics are insufficient to support their key conclusion. There is no experimental counterpart for the predicted upper branch of the spin-split band. The application of Laplacian filtering does not result in visualizing the upper branch, generating a discrete spotty distribution as shown in Fig. 4c. It is recommended to add line profiles of energy and moment distributions to show intensity peaks.

We agree that the upper branch of the altermagnetically split band is difficult to identify in the ARPES raw data. However, we have now added the **new Figure S3** (panel a) to the Supplementary Information, where around the Gamma-point the upper branch is directly visible already in the raw data. As you recommended, we now show line profiles of the ARPES raw data intensity (panel c). We added the following text to the Supplementary Information:

“The altermagnetically split bands appear only faint in the raw data. However, they are directly visible in the symmetrized ARPES intensity (raw data) shown in Fig. S3a. Panel c shows line profiles of the intensity at k-values indicated by the yellow lines in panel a. The upper branch of the altermagnetically split band shows up as a shoulder feature in these profiles. Consistent with the altermagnetic band structure, this shoulder feature is absent for the profiles obtained at the Q and P points. A side-by-side comparison of panels a and b confirms that the application of the Laplace filter emphasizes the bands correctly. The “spots” in the filtered image correspond to crossing points of bands, which result in an increased emission intensity.”

Furthermore, we were able to improve the agreement between the band structure calculations and the experimental ARPES results by considering strain effects on the band structure. Based on a strain analysis of our epitaxial thin films we have now added an additional band structure calculation using the experimentally determined modified bulk lattice parameters of our CrSb thin films (please see our reply to reviewer 1). With these parameters, the calculated altermagnetic band splitting shows an improved agreement with the experimental result (see **new Figure 4d** in the main manuscript).

To clarify this, we added to the manuscript (p. 5):

The experimentally identified upper branch is positioned approximately 200 meV lower in energy compared to the band structure calculated assuming the published hexagonal lattice parameters ($a = 4.123 \text{ \AA}$, $c = 5.47 \text{ \AA}$ [24]). However, our thin films feature slightly different lattice parameters of $a = 4.1 \text{ \AA}$ and $c = 5.58 \text{ \AA}$ (see Supplementary Information). Panel d shows that with these parameters the calculated upper branch of the altermagnetically split band is pushed towards the lower branch resulting in an improved agreement with the experimental data. With such a strong effect of strain on the upper branch, the remaining discrepancies between theory and experiment could be explained by additional small lattice distortions in the surface region of the CrSb thin film mainly probed by SX-ARPES (top 1-2 nm).”

The reasons for the ambiguous data might be the inherent low momentum and energy resolutions of soft-X-ray ARPES and contaminants deposited during sample transportation using the suitcase. It is unclear whether the sample was annealed after installation into the ARPES chamber.

The sample was not annealed after the installation into the ARPES chamber. However, we also investigated a CrSb thin films protected by a 2 nm Si₃N₄ capping layer (also transported in the vacuum suitcase). Both samples gave the same results, only the ARPES intensity from the capped sample was reduced. Thus, we do not believe that surface degradation is the main limiting factor regarding the sample quality. We added the following text to the Supplementary Information: “One reason for the observed band broadening in the experimental data is the mosaicity of the epitaxial thin films shown in Fig. S1c, with a rocking curve width of $\approx 0.5^\circ$ corresponding to $\Delta k_{\parallel} \approx 0.1 \text{ \AA}^{-1}$ for 900 eV photons. Another possible contribution to broadening effects both in energy as well as in momentum could be a relatively small size of sample regions with fully single crystalline order, specifically including the coherent local order of the Sb environment of the Cr sublattices. As swapping the local environment swaps the CEF at the Cr sublattices, this is expected to specifically show up in the characteristic altermagnetic band structure features.”

Spin-resolved band dispersions are necessary for the direct observation of spin-splitting bands of the altermagnet.

Obtaining spin-resolved band dispersion represents the next level of evidence for altermagnetic materials and a more detailed study of the spin polarization, which is beyond the current manuscript, is highly desirable. However, in the process of providing experimental evidence for altermagnets, the observation of band dispersions, which are in agreement with band structure calculations implying the spin splitting, is very significant. To clarify this issue already in the introduction, we added to the manuscript (p. 2):

“We employ spin-integrated soft X-ray angular-resolved photoelectron spectroscopy (SX-ARPES) to probe these bands. Though we do not detect the spin, the observation of band dispersions, which are in agreement with band structure calculations implying spin splitting, provides strong evidence for an altermagnetic band structure of CrSb.”

Such evidence is essential for the stimulation of further research in this field, as it demonstrates the validity of the basic theoretical concepts.

Making use of the spin polarization of altermagnets (including its direct observation by Spin-ARPES) will involve long-term materials growth optimization challenges. In general, macroscopic areas of perfectly single crystalline thin film samples are required, as otherwise the spin polarization will average to zero. Regarding this issue, we added to the Supplementary Information:

“... local order of the Sb environment of the Cr sublattices... This in general to be expected effect does also result in a swapping of the spin polarization of the altermagnetically split bands, even within a single magnetic domain. Thus, it presents a major challenge for potential future spin-resolved ARPES investigations, as averaging over such growth domains will result in averaging the spin polarization to zero.”

The discussion about orbital characters in splitting bands are missing though the authors measured both p- and s-polarized cases.

We are sorry that we forgot to refer the reader in the main text to the Supplementary Information, where the selection rules for photoemission along the Q-P path are discussed. We have now added

to the main text:

p. 3: “For the low-symmetry Q-P path (panels c and d), the situation is more complex. Here, a pronounced polarization dependence, **which is discussed in the Supplementary Information in the framework of selection rules**, and a forward-backward scattering asymmetry inherent to the ARPES geometry (see Methods) is discernible.”

p. 5: “Conversely, when employing s-polarized photons, as displayed in panel b, this lower branch is notably absent, indicating the presence of an essentially even-parity state with respect to the scattering plane as discussed in detail in the Supplementary Information. **In Fig. S5, we show that the altermagnetically split band has mainly the character of a Cr dx^2-y^2 orbital.**”

We now also clarify in the main text that the good agreement between the calculated spectral weight of the Cr-d orbitals agrees well the observed photon polarization dependence of the ARPES intensity, which provides additional evidence for the ability of the calculations to describe our CrSb samples:

p. 5: “In summary, we have combined experimental SX-ARPES investigations with first-principles calculations to identify the altermagnetic band structure of CrSb. Good agreement was obtained, **both regarding the course of the dispersion branches as well as the polarization dependence of the ARPES intensity.**”

Furthermore, we now added an explicit discussion of the Cr vs. Sb character of the electronic states to the Supplementary Information, which includes the **new Figure S4** and the following text:

“Orbital character of the valence states

To identify the orbital character of the electronic bands showing the altermagnetic splitting, we show in Fig. S4a the projections on the Cr as well as on the Sb atomic orbitals. The dominant Cr character is obvious from the comparison of the calculated spectral weights of the orbitals. This result is fully consistent with the experimentally observed periodicity of the ARPES intensity from BZ to BZ, as discussed in the main text. Furthermore, we show in panel b the projections of the electronic bands on the s-, p-, and d-orbitals of Cr. We observe, that the major contribution originates from d-orbitals, which we discuss below in the framework of selection rules in photoemission spectroscopy.”

Soft x-ray ARPES can often be utilized, and band-dispersion results are usually plotted with the calculations. A similar observation was reported for RuO₂ using the MCD-ARPES technique.

<https://arxiv.org/abs/2306.02170>

While SX-ARPES is a standard method, preparing altermagnetic samples which enable the use of this method is a major challenge. We are only aware of the references [23-25] reporting direct observation of altermagnetic band splitting (in MnTe).

The reference you mention above corresponds to our Ref. [22]. This reference reports the observation of a magnetic circular dichroism in RuO₂, which does not appear in conventional antiferromagnets and is interpreted as evidence for an altermagnetic band structure. However, the band splitting is not directly observed in this work. Thus, this very interesting work is complementary to our results as it investigates a different material and provides a different type of evidence for an altermagnetic band structure.

We hope that our remarks above and specifically the new ARPES data evaluation and calculations of the orbital character of the CrSb electronic states have convinced you that our work represents a substantial progress in the evolving field of altermagnetism.

REVIEWER COMMENTS

Reviewer #1 (Remarks to the Author):

The author has responded to my question as per my request, and I believe that the article has reached the level of publication now.

- 1, In SUPPLEMENTARY INFORMATION Fig. 4, why lost the lines for “Sb (right) atomic orbitals and Projection of the Cr s- (left), p- (center)” in the figures?
- 2, We are not aware of any observation of an AHE in CrSb.? They should check this again especially recently reported related this material.

Reviewer #2 (Remarks to the Author):

The reply of the authors is satisfactory except for the last question. As many of the authors of the paper know very well, the anomalous Hall effect is a property of the altermagnets. Once the Néel vector is out of the plane, you can have ρ_{xz} and/or ρ_{yz} different from zero. The theoretical proposal of AHE for CrSb is presented here <https://arxiv.org/abs/2312.07678> . Anyway, the AHE was not the main goal of the paper.

I recommend the paper for publication in Nature Communication.

Reviewer #3 (Remarks to the Author):

The authors have improved the manuscript based on the reviewers' comments. However, there are a few remaining points I want to point out. The authors claim to have recalculated the band structures using the lattice parameters from the xrd data and obtained results in Fig. 4d that agree with the ARPES experiment. But when we look at Fig. 4d, the red lines of band branches near the center and the upper right side of the panel do not have experimental counterparts. The authors do not give any clear and rational explanation. Could additional small lattice distortions as described by the authors cause these discrepancies? Can some lattice distortions be found on the TEM images? Also, why are the red and blue lines of the bands in Figures 4c and 4d reversed? In addition, I understand that spin-resolved band dispersion is a bit difficult to do, but the fact remains that the authors do not present it as direct evidence for the observation of altermagnetic band splitting as stated in the title of the manuscript. Based on the above, I am currently neutral on the publication of this manuscript in Nature Communications.

We would like to thank the reviewers for their insightful comments regarding our manuscript, which helped us to improve it considerably.

Please find below our point-by-point response to all reviewer's concerns (3 pages). For improved readability, citations from the reviewer reports are printed in black, our reply is printed in grey, and new text added to the manuscript is printed in blue (here and in the manuscript).

Reviewer #1 (Remarks to the Author):

The author has responded to my question as per my request, and I believe that the article has reached the level of publication now.

1, In SUPPLEMENTARY INFORMATION Fig. 4, why lost the lines for "Sb (right) atomic orbitals and Projection of the Cr s- (left), p- (center)" in the figures?

The color scale of these plots indicates the spectral weight. The fact that the lines are almost white (i.e. almost lost) indicates that the spectral weight is almost zero. This means that the Sb orbitals and the Cr s- and p-orbitals do not contribute to the ARPES intensity. To make this more obvious to the reader, we added to the caption of Fig. S4:

"The color scale indicates the spectral weight, i. e. bands with low spectral weight appear faint in the plots."

2, We are not aware of any observation of an AHE in CrSb.? They should check this again especially recently reported related this material.

We checked the available literature again and could not find any report of an AHE observation for CrSb. An AHE was observed for MnTe [20,21] and this compound has the same crystal structure as CrSb, but a different orientation of the magnetic moments. However, the orientation of the magnetic moments is essential for the existence of an AHE. E.g., in the supplementary material of Ref. [21], it is shown that in hexagonal MnTe, an AHE is possible only for the Néel vector N along a $\langle 110 \rangle$ direction, but not for $N \parallel \langle 100 \rangle$. Please note that in CrSb the easy axis is parallel $\langle 001 \rangle$, in which case we do not expect an AHE. How the Néel vector of CrSb should be rotated out of the c-axis direction is not clear. For magnetic fields smaller than 14 T we could not observe any indication of a spin-flop transition, which would also need to result in $N \parallel \langle 110 \rangle$, not in $N \parallel \langle 100 \rangle$, in order to be associated with an AHE.

Reviewer #2 (Remarks to the Author):

The reply of the authors is satisfactory except for the last question. As many of the authors of the paper know very well, the anomalous Hall effect is a property of the altermagnets. Once the Néel vector is out of the plane, you can have ρ_{xz} and/or ρ_{yz} different from zero. The theoretical proposal of AHE for CrSb is presented here <https://arxiv.org/abs/2312.07678>. Anyway, the AHE was not the main goal of the paper.

We would like to thank the reviewer for drawing our attention to the above-mentioned manuscript. However, we do not see any indication of an AHE in CrSb, as described above in our reply regarding the second issue discussed by reviewer 1.

I recommend the paper for publication in Nature Communication.

Reviewer #3 (Remarks to the Author):

The authors have improved the manuscript based on the reviewers' comments. However, there are a few remaining points I want to point out. The authors claim to have recalculated the band structures using the lattice parameters from the xrd data and obtained results in Fig. 4d that agree with the ARPES experiment. But when we look at Fig. 4d, the red lines of band branches near the center and the upper right side of the panel do not have experimental counterparts.

While the lower branch of the alternatingly split band agrees very well with the ARPES intensity distribution, the calculated upper branch is positioned 100 - 200 meV above the main ARPES intensity. Nevertheless, the dispersion of the ARPES intensity agrees qualitatively with the calculated upper branch. Near the center of the BZ, the experimental dispersion is just a little less steep than the theoretical prediction and even in the upper right region, although there the ARPES intensity is only faint, similar dispersions are observed.

The authors do not give any clear and rational explanation. Could additional small lattice distortions as described by the authors cause these discrepancies? Can some lattice distortions be found on the TEM images?

Analyzing the atomic positions in our TEM data (Fig. S2,a and additional images), focusing on the bulk of the thin films, we found evidence for a tiny lattice distortion within the hexagonal plane as indicated on the right. We calculated the BS for this slightly non-hexagonal cell, but the result is almost indistinguishable from the hexagonal cell calculation shown in Fig. 4d.

Furthermore, we analyzed the TEM image showing the surface/ interface region of the CrSb/Si₃N₄ thin film (Fig. S2,c), which provides evidence for additional lattice distortions in this region. We added the following text to the Supplementary Information (p. 2):

Evaluating intensity line profiles through columns of atoms oriented perpendicular to the CrSb/Si₃N₄ interface, we identified a reduced distance (5-15%) between the last two Sb layers at the interface compared to the deeper layers. Such lattice distortions in the surface region could explain the discrepancies remaining in Fig. 4d (of the main manuscript) between the band structure calculations and the ARPES results.

This statement is based on the data shown in the figure on the right. The yellow box in the TEM image shows one of ten regions, from which intensity profiles were obtained. The sum of the intensity profiles and the manually extracted positions of the atoms in arbitrary units (pixels) are shown in the graph. This data shows the shrinking of the Sb-Sb layer distance approaching the interface with the Si₃N₄ capping layer.

Also, why are the red and blue lines of the bands in Figures 4c and 4d reversed?

Thank you for pointing this out. We swapped now the red and blue color of the calculated bands in Fig. 4d to obtain a consistent representation.

In addition, I understand that spin-resolved band dispersion is a bit difficult to do, but the fact remains that the authors do not present it as direct evidence for the observation of altermagnetic band splitting as stated in the title of the manuscript. Based on the above, I am currently neutral on the publication of this manuscript in Nature Communications.

We do not probe the spin splitting, but we observe the band splitting in the altermagnetic band structure directly. Regarding the title, some shortening was required to keep it readable. In the abstract of our manuscript, we explain that our measurements are spin integrated. By “direct evidence”, we want to point out that we observe the electronic bands, which are expected to be split in an altermagnet, directly. This is in contrast to measurements of electric transport properties, which by the observation of characteristic behaviour represents more indirect evidence for altermagnetism than our overall good agreement of the ARPES results with the spinintegrated band structure.

Please note, that up not know only for MnTe ARPES results showing split bands directly are available (Refs. 23,25,26) and from those only Ref. 25 probed spin polarization obtaining values in the single digit percent range. Thus, we consider our results of sufficient significance for publication in Nature Communications.

REVIEWERS' COMMENTS

Reviewer #3 (Remarks to the Author):

Although I am not so satisfied with the revisions, which cannot fully address my concerns, I have decided to recommend publication anyway. It is so difficult for the authors to reach (1) a quantitative agreement between the dispersion of the ARPES intensity and the calculated upper branch, and (2) a spin-resolved ARPES experiment. It is a timely work on CrSb, so publication of the results, even if not perfect, is certainly warranted.